# Cellular Senescence, Aging and Non-Aging Processes in Calcified Aortic Valve Stenosis: From Bench-Side to Bedside

**DOI:** 10.3390/cells11213389

**Published:** 2022-10-27

**Authors:** Andrea Ágnes Molnár, Dorottya Pásztor, Béla Merkely

**Affiliations:** Heart and Vascular Center, Semmelweis University, 1122 Budapest, Hungary

**Keywords:** aging, senescence, aortic valve, calcification, imaging

## Abstract

Aortic valve stenosis (AS) is the most common valvular heart disease. The incidence of AS increases with age, however, a significant proportion of elderly people have no significant AS, indicating that both aging and nonaging pathways are involved in the pathomechanism of AS. Age-related and stress-induced cellular senescence accompanied by further active processes represent the key elements of AS pathomechanism. The early stage of aortic valve degeneration involves dysfunction and disruption of the valvular endothelium due to cellular senescence and mechanical stress on blood flow. These cells are replaced by circulating progenitor cells, but in an age-dependent decelerating manner. When endothelial denudation is no longer replaced by progenitor cells, the path opens for focal lipid deposition, initiating subsequent oxidation, inflammation and micromineralisation. Later stages of AS feature a complex active process with extracellular matrix remodeling, fibrosis and calcification. Echocardiography is the gold standard method for diagnosing aortic valve disease, although computed tomography and cardiac magnetic resonance are useful additional imaging methods. To date, no medical treatment has been proven to halt the progression of AS. Elucidation of differences and similarities between vascular and valvular calcification pathomechanisms may help to find effective medical therapy and reduce the increasing health burden of the disease.

## 1. Introduction

The prevalence of calcified aortic stenosis (AS) is increasing, probably as a consequence of the worldwide aging population. Additionally, a further exponential increase in the elderly population demographics is expected by 2050, which further increases the impact of AS [1,2]. The Tromsø Study assessed the prevalence, incidence, prognosis and progression of aortic stenosis in the general population [3]. The incidence of AS in the study was 4.9‰/year from 1994 to 2008 [3]. Notably, the overall public health burden of AS is escalating rapidly due to the increasing lifespan and prevalence of risk factors [4,5,6]. However, according to the current guidelines, no effective medical treatment is available [7]. The only treatment of AS represents surgical or transcatheter aortic valve replacement with an exponentially growing number of procedures, hence bearing a considerable clinical and economic burden [1]. Accordingly, there is an unmet need for further medical treatment options capable of slowing disease progression. A comprehensive understanding of the initiation and progression pathways of aortic valve calcification is critical. Nonetheless, a significant proportion of elderly people have no significant AS, indicating that both aging and nonaging pathways are involved in the pathomechanism of the disease [8,9,10,11,12,13]. Previously, AS was thought to be a passive, degenerative process involving age-related, replicative cellular senescence [11]. Now it is understood to be a complex process with active elements, including endothelial injury, chronic inflammation, fibrosis, lipid deposition, matrix remodeling and calcium deposition [8,9,10,11,12,13]. Current routine clinical diagnostic tools can identify only the later stages of the disease when calcification is already present [7]. Future molecular and imaging diagnostic tools could help to identify the early stages, initiating the preclinical stage of the disease prior to irreversible macroscopic and later-stage valve calcification [8,13]. This review summarizes the stages of aortic valve calcification, emphasizing the importance of its understanding in order to find effective medical therapy and reduce the increasing health burden of the disease.

## 2. The Structure of Aortic Valve

The normal structure of the aortic valve is avascular with three semilunar cusps and is part of the aortic root connecting the heart to the systematic circulation [14,15]. All components of the aortic root, including the annulus, sinuses of Valsalva, sinotubular junction, interleaflet triangles, commissures and the three aortic valve leaflets, interact with each other to maintain optimal coronary perfusion and unidirectional laminar blood flow through the vascular system [14,16]. The three aortic valve leaflets (or cusps) are named according to the name of the coronary artery derived from the cusp. The right and left coronary cusps are named after the right and left coronary arteries, while the cusp without a deriving coronary artery is determined as a noncoronary cusp. The cusps maintain unidirectional forward blood flow from the left ventricle to the aorta, meanwhile they must be compliant to open and simultaneously must be able to resist the high-pressure environment of the ventricular systole [14,17]. Previous biomechanical studies, comparing aortic valve anatomical structure-induced mechanical strain alterations, showed that the most optimal anatomical configuration is the trileaflet aortic valve [15,18,19]. The leaflet strain increases from the base of the valve to the tip, both in the case of tricuspid and bicuspid aortic valves [15]. However, the geometry of the bicuspid aortic valve increases the overall mechanical stress, mainly at the commissure [15]. The site of increased mechanical stress usually represents the initiating location of aortic valve degeneration. The bicuspid aortic valve is a congenital alteration as a result of the fusion of two cusps. The prevalence of bicuspid aortic valves is 0.5–2% worldwide, with a male predominance [20,21]. According to the location of fusion and the presence of fibrous raphe, several morphological types can be distinguished. The most commonly used Sievers classification differentiates three main types according to the number of raphes. There is no raphe in the case of a type 0 bicuspid aortic valve. Only one raphe is present in the case of the most common type 1, and two raphes in the case of type 2 bicuspid aortic valves [22,23]. The most common right and left coronary cusp fusion is defined as a coronary cusp fusion with a prevalence of 80%. The mixed types of fusions of right and noncoronary cusp fusions or the left and noncoronary cusp fusions are less common (17% and 2%, respectively), however, both of them represent a higher risk factor for the development of aortic stenosis compared to the common coronary cusp fusion.

The aortic valve has an aortic and a ventricular surface side due to its spatial location within the aortic root [10]. Histologically, both sides are covered by valvular endothelial cells (VEC) to ensure a nonthrombogenic surface layer and to regulate inflammatory reactions [10]. Valvular endothelial cells represent a barrier on the surface of the valve between the tissue and blood, similar to vascular endothelial cells. However, these cells have a different phenotype compared to vascular endothelial cells in terms of barrier function, proliferative potential and sheer stress response. Diffusion of oxygen and nutrients through the valvular endothelial layer to the inside of the valve is critical, as the valve is avascular compared to the vascular system’s blood supply through the vasa vasorum. Furthermore, the arrangement of VEC is perpendicular to the direction of blood flow and not parallel with it, compared to the arrangement of vascular endothelial cells [24]. Additionally, the VECs are mechanosensitive that respond to mechanical stimuli [25].

Each aortic cusp consists of three layers: aortic fibrous layer, spongy layer and ventricular layer. The ventricular layer is composed of radially aligned elastin fibers to promote cusp motion, while the aortic fibrous layer consists of circumferentially aligned collagen fibers [10,15]. The proteoglycan-rich spongiosa layer can be found between them, encompassing mainly glycosaminoglycans to offer lubrication from shear forces [10,15]. The valvular interstitial cells (VIC) are quiescent fibroblast-like cells found throughout the three layers, producing and repairing the extracellular matrix over time [10,15]. Although VICs are considered to be a fibroblast-like population, they have substantial phenotypic plasticity [10]. The interplay between the cellular and extracellular matrix components of the aortic valve forms an integrated response to the mechanical effects of different hemodynamic situations to maintain normal aortic valve functions [10,15].

## 3. Pathomechanism of Aortic Valve Calcification: Senescence and Steps of Calcification

### 3.1. The Role of Cellular Senescence in Aortic Valve Calcification

Cellular senescence is a feature of somatic cells defined by a nondividing, irreversible cell cycle arrest state due to intrinsic and/or extrinsic factors [26]. The loss of replicative capacity is a consequence of replication-related telomere shortening or mechanical and metabolic stressors leading to deoxyribonucleic acid (DNA) damage, mitochondrial dysfunction and the accumulation of reactive oxygen or nitrogen species [26,27,28,29]. It is worth noting that cellular senescence is different from cellular quiescence, which is an adaptive response to the nutrient environment resulting in reversible cell cycle arrest [26]. Telomeres are nucleoprotein complexes at the cap of the chromosomes with tandem repeats of DNA and a six-protein complex called shelterin [30]. This cap of the chromosome is largely double-stranded, however, it ends in a short single-strand, resulting in protection and replication difficulties at the end of the chromosome [30]. The shelterin protein complex binds to the double- and single-stranded telomere DNA and protects it from unwanted degradation [30]. Without shelterin proteins, the ends of chromosomes could be misrecognized by the DNA damage response and repair machinery as double-strand breaks, which require repair. It is known that DNA polymerase is unable to fully replicate chromosome ends, consequently, it shortens with each replication cell cycle due to DNA loss [31]. The progressive telomere shortening results in a critical telomere length with each cell division and the somatic cell becomes senescent, also termed as a replicative senescent cell [32]. To counteract replication difficulties, the telomerase enzyme can synthesize new telomere repeats at chromosome ends, however, it is silent in most somatic tissues and is only expressed in germline cells and a subset of proliferating somatic progenitor cells [30,32,33]. Thus, normal somatic cells become senescent when their telomeres reach the threshold length.

Immune cells can remove age-related, replicative senescent cells and prevent their accumulation, except in the case of increased stress-induced cellular senescence and immune system dysfunction [27]. Nonetheless, senescent cells are not inactive cells, as they secrete cytokines, chemokines and matrix metalloproteinases defined as senescence-associated secretory phenotypes (SASP). This contributes to the extracellular matrix remodeling and valvular structural changes associated with aging. The age-related structural changes in the aortic valve include increased collagen content and crosslinking, leading to increased leaflet stiffness, which is adopted for age-related physiologic changes in cardiac hemodynamics in order to maintain the normal aortic valve function [34,35]. Apart from physiologic, age-related senescence and further pathologic cellular senescence can be induced by cellular stressors, such as excessive mechanical stress, oxidative stress, metabolic stress, and factors leading to DNA damage, also termed stress-induced premature cellular senescence [26,27]. Chronic excessive stress over the years may result in the accumulation of pathologic senescent endothelial cells, mainly on the aortic side of the valve, where the blood flow is oscillatory. Notably, aging of the immune system, defined as immunosenescence, may result in decreased clearance and accumulation of senescent cells [27]. However, aging rarely leads to severe aortic valve stenosis, as a significant proportion of the elderly population does not develop significant AS. In addition to aging, excessive mechanical stress, genetic factors and metabolic factors, such as high blood pressure, overweight and hypercholesterinemia, can induce and aggravate pathological cell senescence and calcification [36,37]. Excessive mechanical stress is present in the case of increasing aortic stenosis severity when the blood flow is oscillatory on the aortic side and turbulent on the ventricular side. Oscillatory flow represents a different mechanical stress compared to the laminar flow of the normal aortic valve. Oscillatory shear stress has been shown to promote atherosclerotic plaque formation in arteries, and this is also assumed to be the case in aortic valve calcification, as calcification is typically present on the aortic side of the valve [10]. Other than senescence, mechanical stress can lead to focal endothelial damage and denudation [38].

Mechanical stress induced endothelial denudation and focal tissue disruption is repaired either by activated, somatic quiescent endothelial cell division or circulating endothelially progenitor cell adhesion to the damaged site [38]. However, the turnover of these perilesional activated endothelial cells is low, and endothelial progenitor cells are needed to facilitate endothelial repair. The circulating endothelial progenitor cells are bone marrow-derived cells, which can divide into somatic valvular endothelial cells [38]. Accumulation of senescent endothelial cells diminishes the regeneration of endothelial disruption, as these cells cannot divide. Furthermore, aging affects the regenerative capacity of endothelial progenitor cells as the release of these cells from the bone marrow decreases as age increases [38]. Moreover, the senescence of endothelial progenitor cells increases as age increases. Matsumoto and coworkers showed enhanced apoptosis as well as increased senescence of circulating endothelial progenitor cells in patients with aortic stenosis, leading to a reduced circulating number of these cells. It is proposed that besides aging, cardiovascular risk factors, such as hypertension, diabetes, hyperlipidemia and smoking can influence the number of circulating endothelial progenitor cells and the regenerative capacity of the cardiovascular system, including the vascular system and the valves [39,40]. Molecular markers of cellular senescence, such as beta-galactosidase and cell cycle arrest inductor P16INK4A (inhibitor of cyclin D-dependent kinases), have shown a correlation with tissue remodeling severity and degenerative changes in the aortic valve [37].

In conclusion, senescent cells contribute to aging and aortic valve degeneration not only as a result of cell cycle exit and resistance to apoptosis, but also by secretion of pro-inflammatory cytokines, chemokines, matrix metalloproteinases and growth factors, promoting senescence in surrounding cells as a bystander effect [41,42]. Eliminating senescent cells might represent a future therapeutic strategy. However, human applications of these potential therapies are still limited by our sparse knowledge of the basic molecular cell biology of senescence [43].

### 3.2. The Two Phases of Aortic Valve Calcification: Initiation and Progression Phase

Previously, it was thought that aortic valve sclerosis was a passive process as a consequence of aging. It has already been revealed that aortic valve sclerosis is an active process with some similarities and differences compared to vascular atherosclerosis. Aortic valve sclerosis and stenosis are different stages of aortic valve calcification.

The early phase of aortic valve degeneration usually begins at the aortic side with the dysfunction of the endothelial barrier, allowing lipids from the blood to enter into the subendothelial space [44,45,46]. Valvular endothelium disruption may occur as a result of several aging and nonaging factors, usually as a multifactorial and complex process. Age-related (replicative) and stress-induced cellular senescence processes are discussed above, and further proinflammatory and profibrotic processes are involved in the initiation phase [8,9,10,11,12,13]. Even under physiological circumstances, the mechanical stress pattern caused by the blood flow over the years might initiate aortic valve sclerosis, affecting mainly the aortic side of the valve, usually beginning at the base of the leaflet [47]. On the aortic side of the normal valve, within the sinuses of Valsalva, valvular endothelial cells are exposed to “sclerosis prone” oscillatory low shear stress in systole and turbulent flow vortices in diastole. Meanwhile, cells on the ventricular side of the normal aortic valve experience “less sclerosis prone” linear high-shear stress of systolic forward laminar flow [9,48]. The laminar flow on the ventricular side becomes turbulent in the case of AS. Cheng C. and coworkers worked out a shear-stress carotid artery mouse model to examine plaque formation under low, high and low-oscillatory shear stresses [49]. Atherosclerotic lesions evolved in the regions of low shear stress in this mouse model [49]. In this region, the expression of proatherogenic inflammatory mediators and matrix metalloproteinase activity was higher [49]. Expression of vascular cell adhesion molecule-1 (VCAM-1), intercellular adhesion molecule-1, vascular endothelial growth factor, C-reactive protein and proinflammatory cytokine interleukin 6 (IL-6) was upregulated in the lowered shear stress region [49]. It is worth noting that the physiologic shear stress pattern of laminar flow over time might not result in significant AS, as a significant proportion of the elderly do not suffer from significant AS [9]. Additional genetic and acquired factors may have additive effects on the development of significant AS [50]. Furthermore, the congenital bicuspid aortic valve morphology represents an altered mechanical stress, as the systolic oscillatory shear stress and the ascending aorta wall shear stress are altered compared to the tricuspid aortic valve. Subsequently, the onset of aortic valve degeneration is earlier in the case of bicuspid aortic valve, the progression of the disease is more rapid and, in many cases accompanied by the dilatation of the ascending aorta [51]. Further nonaging factors, such as radiation, have been revealed to initiate valvular endothelial injury and valvular inflammation, leading to aortic valve disease [52].

Endothelial dysfunction and disruption in the early phase lead to lipid deposition from the blood into the subendothelial space. Moreover, lipid deposition and endothelial barrier dysfunction initiate inflammatory cells and cytokines to enter into the valvular interstitial space [9,10,11]. These inflammatory cytokines (tumor necrosis factor-α, interleukin-6 and interleukin-8) promote endothelial to mesenchymal transformation, resulting in a new myofibroblastic cell phenotype migrating into the interstitial space [53,54,55,56]. The endothelial to mesenchymal transition was first described in 2001 by Paranya and coworkers, however, there is still debate regarding the role of the activated myofibroblast-like VEC in the extracellular matrix regulation of the aortic valve [10,53,54]. The membrane of a VEC encompasses the endothelial nitric oxide synthase (eNOS) to produce nitric oxide (NO), which inhibits fibrosis and calcification [57,58]. Valvular endothelial NO increases the expression of neurogenic locus notch homolog protein 1 (NOTCH1) signaling in VIC, which inhibits regulators of osteoblast cell fate (e.g., runt-related transcription factor 2, RUNX2) and increases the expression of anticalcification factors, such as SRY-Box transcription factor 9 (SOX9) and bone morphogenic protein 2 (BMP2) [59]. This NOTCH1-RUNX2-SOX9-BMP2 signal route is short-defined as the NOTCH1 pathway. Aging is associated with increased oxidative stress, decreased extracellular superoxide dismutase activity and decreased eNOS activity leading to a diminished NO bioavailability and endothelial dysfunction [57,58]. Valvular endothelial cell dysfunction and the subsequent reduced NO production promote a fibrotic process within the valve. Furthermore, the upregulated renin-angiotensin system (RAS) is also critical in aortic valve disease [11]. Valvular endothelial cells under increased shear stress promotes TGFβ1 to activate the quiniscent VIC set in the fibrosa, spongiosa and ventricularis layers [60]. The activation of these cells may result in myofibroblastic differentiation characterized by the expression of alpha-smooth muscle actin (αSMA). The myofibroblastic VICs secrete structural matrix proteins and matrix metalloproteinases leading to extracellular matrix remodeling, leaflet thickening and increased leaflet stiffness [61]. Microcalcification, also defined as dystrophic calcification, in the early phase, is a result of myofibroblastic VIC death and the release of apoptotic bodies in the area of lipid deposition and inflammation [11]. Briefly, under physiological shear stress circumstances, VEC protects VIC from myofibroblastic differentiation by reducing αSMA expression, and calcification by producing NO and increasing the expression of NOTCH signaling target genes in these cells [59,62]. Moreover, VIC can suppress endothelial-to-mesenchymal transformation and osteogenic differentiation of VEC, emphasizing the importance of VEC and VIC interactions in valve homeostasis [63].

In the progression phase of the disease, there is a constant remodeling of the extracellular matrix and ongoing calcification, leading to impaired leaflet opening and closing over the cardiac cycle. Previous in vitro studies revealed osteoblastic differentiation of quiniscent VIC by exposure to BMP2, RUNX2 and osteopontin [64]. Moreover, Schloter and coworkers showed that myofibrotic differentiation may precede osteoblastic differentiation of VIC. This was demonstrated with transcriptomics, showing that VIC isolated from fibrotic areas of valves exhibited intermediate gene profiles between nondiseased and calcific regions [65]. Osteogenic differentiation factors implicated in osteogenic cell differentiation are upregulated, including the NOTCH pathway, receptor activator of nuclear kappa B (RANK)-RANK ligand (RANKL) and osteoprotegerin (OPG) pathway [11]. Dayawansa and coworkers proposed that progression of aortic valve stenosis promotes further mechanical stress alterations, endorsing a positive feedback loop mechanism and a vicious cycle of chronic inflammation and calcification [9]. Once calcification develops in the valve, a constant vicious circle of calcification and valve injury is maintained. Calcific deposits in the leaflets lead to increased mechanical stress and injury-induced activation of further osteoblast differentiation [11].

Overall, the early phase of aortic valve degeneration is dominated by inflammation, subendothelial lipoprotein oxidation, fibrosis and microcalcification, while the later phase is dominated by self-perpetuating progressive calcification. Valvular interstitial cell activation with myofibroblast differentiation is present mainly in the early phase, while the osteoblast differentiation is mainly in the later stage. Grim and coworkers showed that inflammatory macrophages may initiate a myofibroblast-to-osteogenic intermediate VIC phenotype, which may mediate the switch from fibrosis to calcification during AS progression [66]. Nevertheless, the early phase is dominated by interleukins secreted by macrophages, whereas the later phase is dominated by the NOTCH and RANK/RANKL/OPG pathways [11,55,56]. Focal discrete aortic valve calcification is defined as aortic sclerosis. Approximately 10–15% of patients with aortic valve sclerosis will progress to obstructive calcification in their lifetime with mild, moderate or severe stenosis [67]. Furthermore, AS is not only a valvular disease, as the increased afterload induces adaptive hypertrophic left ventricular remodeling and may lead to heart failure. Most patients enter the healthcare system because of heart failure symptoms, hence the valve disease is diagnosed in this late phase [68].

## 4. Risk Factors of Aortic Valve Calcification

Previous studies have shown that age, obesity, smoking, coffee intake, hypertension, diabetes, kidney disease and bicuspid aortic valve are associated with an increased risk of aortic valve stenosis [69,70,71,72,73,74,75,76]. The Cardiovascular Health Study is a large-scale prospective, longitudinal, population-based study of the elderly, aimed at determining risk factors of aortic sclerosis and stenosis [69]. Age, male gender, smoking, hypertension, height, high lipoprotein(a) (Lp(a)) and low-density lipoprotein cholesterol (LDL-C) levels were found as main risk factors associated with aortic valve degeneration in the study [69]. In the Cardiovascular Health in Ambulatory Care Research Team (CANHEART) population-based observational study, 1.12 million individuals above the age of 65 years were followed for a median of 13 years, of which 20,995 subjects developed severe aortic stenosis [77]. Hypertension, diabetes, dyslipidemia and a combination of these factors were found to be independent and additive risk factors of severe aortic stenosis, of which hypertension had the highest attributed risk because of its higher prevalence in the elderly [77]. Upregulation of the renin-angiotensin system (RAS) is critical in the development of hypertension and also aortic valve disease, as it might impose increased mechanical stress on the valve [10]. Moreover, in the Multi-Ethnic Study of atherosclerosis (MESA), different stages of hypertension showed a different prevalence of aortic valve sclerosis. Calcification was observed in 6% of normotensive individuals, in 11% of borderline hypertensive individuals, in 17% of individuals with stage I hypertension and in 16% of individuals with stage II hypertension [78]. In the PROGRESSA study, a more rapid progression of aortic valve stenosis could be observed in the presence of hypertension [79].

Despite the fact that aortic valve degeneration is not exclusively attributable to aging, age remains a strong predictor, as the risk for aortic valve calcification doubles every 10 years above the age of 65 years [69,78,80]. Nevertheless, age was not associated with the rate of aortic disease progression [78,81]. In the Cardiovascular Health Study and Framingham Offspring Study, the higher levels of LDL-C and total cholesterol were associated with a greater probability of developing aortic sclerosis [69,82]. In the early phase, apolipoprotein (apo) B-containing lipoproteins accumulate in the subendothelium, including very-low-density, intermediate-density and low-density lipoprotein particles, as well as Lp(a) [68]. Despite the crucial role of lipids in the initiation phase, their late phase effect is not negligible, as patients with elevated Lp(a) levels demonstrate faster disease progression even in the propagation phase [68]. However, lipid-lowering medical therapy proved to be ineffective in delaying or reversing the progression of aortic valve calcification. This suggests that lipids may exert a larger influence on the aortic valve disease and that the initiation of lipid-lowering therapy at the time of disease diagnosis is too late [80,83,84]. In the CANHEART study, diabetes mellitus and adiposity were also associated with a higher risk of aortic stenosis [77]. Chronic kidney failure has been linked to a higher incidence and progression of aortic calcification [70]. The coincidence of aortic valve calcium and coronary artery calcium is a common complication in end-stage renal disease [85]. Dai and coworkers demonstrated that aortic valve sclerosis increased all-cause mortality independently of the presence of coronary artery calcification, traditional cardiovascular risk factors and inflammation [85]. Coronary artery disease is common in patients with aortic stenosis [86]. In the MESA study, coronary artery disease was present in 82% of the aortic valve calcification group, while 45% in participants without aortic valve disease [87].

In general, the rate of disease progression from aortic sclerosis to any severity of stenosis is low (<2% per year), however, it depends on additional risk factors [88]. It is worth noting that the rate of aortic valve calcification is variable. More rapid progression could be detected in elderly men associated with coronary artery disease or in the case of smoking, hypercholesterolemia and elevated serum creatinine levels [88]. The bicuspid aortic valve represents a higher risk of aortic valve degeneration, with an earlier onset and a more rapid progression. The presence of only two leaflets increases mechanical stress and initiates tissue remodeling in the raphe region [71]. It is an autosomal-dominant disease with several gene mutations, including the NOTCH1 mutation and GATA family mutations (transcription factors characterized as zinc finger proteins that bind the consensus DNA sequence (T/A)GATA(A/G)) [71,89]. Furthermore, large-scale genome-wide associated studies identified a variant in the palmdelphin (PALMD) and fatty acid desaturase 1/2 (FADS1/2) locus linked to aortic stenosis and bicuspid aortic valve [90,91,92]. PALMD is supposed to impact VIC differentiation and FADS1/2 fatty acid biosynthesis.

These data implicate that vascular atherosclerosis and aortic valve sclerosis share mostly the same risk factors, supporting many similarities between the pathogenesis of early phase aortic valve disease and atherosclerosis [69]. Unlike vascular atherosclerosis, however, modification of these factors does not significantly alter the mortality associated with aortic stenosis.

## 5. Similarities and Differences between Aortic Valve Sclerosis and Vascular Atherosclerosis

The pathomechanism of aortic valve sclerosis and vascular atherosclerosis share many similarities in the initiation phase, however, the progression phase is much different [38,46,93]. Endothelial dysfunction, inflammation, lipid infiltration and microcalcification are common in both diseases in the early phase, however, some studies shed light on differences even in the early phase between the two diseases. Olsson and coworkers showed that lipids remain at the subendothelial layer in the aortic valve while they infiltrate deeper in the case of arteries [45]. Lipid deposition initiates the accumulation of inflammatory cells, including macrophages, in both types of tissue, however, they form high-density foam cells and necrotic cores only in the case of vascular atherosclerotic plaque [44,45]. Vascular smooth cells form a fibrous cap over the atherosclerotic cap and prevent thrombus formation until plaque rupture. Inflammation and lipid infiltration are less important in the later phase, as the mechanical stress and the interaction between inflammatory cells and calcification mediators become crucial in the progression of aortic valve disease [93]. Notably, the unstable process with plaque rupture leading to acute major adverse events is a feature of vascular atherosclerosis, meanwhile aortic valve calcification is a stable progressive process with lamellar bone formation [93]. Furthermore, the impact of cellular senescence also differs between aortic valve stenosis and vascular atherosclerosis. In the case of vascular atherosclerosis, senescent cells secrete SASP factors, promoting plaque instability and the progression of atherosclerosis [94]. However, by inhibiting monocyte and macrophage proliferation, these cells can also limit the growth of atherosclerotic plaques [94]. Senescent cells of the aortic valve secrete SASP factors, leading to age-related extracellular matrix remodeling and increased leaflet stiffness, which nevertheless are needed to develop stenosis. Multiple atherosclerosis risk factors, such as age, male gender and high blood pressure, are associated with aortic valve sclerosis as well [95]. However, modification of these factors in the case of aortic stenosis did not substantially decrease the mortality risk.

## 6. The Imaging of Aortic Valve Degeneration in Clinical Practice

Transthoracic echocardiography (TTE) is the gold standard method for the diagnosis of aortic valve stenosis. The aortic valve cusps become thickened and calcified progressively as the severity of the disease increases and the extent of leaflet systolic opening and diastolic closure decreases. The normal aortic valve opening area in adults is 3–4 cm^2^. The initiation and early phase of the disease usually does not impact the opening area and is difficult to detect in the clinical routine evaluation. The area can decrease in the later phase of calcification, resulting in different stages of severity. In the early course, the aortic valve opening area progresses with minor changes, usually with a 0.1 cm^2^/year decrease, however, it can be variable depending on risk factors [35,96]. The aortic valve stenosis is considered severe when the opening area decreases to 1 cm^2^ or less. Furthermore, the stenotic valve leads to flow acceleration and pressure gradient elevation between the left ventricle and the ascending aorta.

The clinical standard echocardiographic parameters used to quantify the severity of aortic stenosis include the peak aortic jet velocity, the mean pressure gradient, and the aortic valve opening area measurement [97]. Aortic valve opening area can be measured either by the continuity equation (functional area) or by direct tracing (anatomical area), both bearing their own limitations [97]. The continuity equation is based on the law of conservation of mass using two-dimensional (2D) and doppler echocardiography, meaning that the flow volumes proximal to the stenotic valve are equal to the flow volumes distal to the aortic valve [97]. Flow volume measurements need diameter measurements of the left ventricular outflow tract (LVOT) and blood flow velocity measurements at the level of LVOT and valve stenosis. However, this hemodynamic measurement holds some possible errors, including LVOT diameter measurement, as it is more elliptical than circular [97]. Usually, we measure the shorter anteroposterior diameter of the left ventricular outflow tract with transthoracic echocardiography. Furthermore, the pressure gradient depends on stroke volume and cardiac output. High cardiac output states, such as fever, hyperthyroidism, anemia, dialysis and aortic regurgitation, may lead to high gradients despite nonsignificant aortic valve stenosis. Nonetheless, low gradients may occur despite significant stenosis in the case of severe left ventricular systolic dysfunction, mitral valve regurgitation and left to right intracardiac shunts [97]. Another challenging situation occurs when aortic valve stenosis and hypertrophic obstructive cardiomyopathy coexist. In this case, valvular and subvalvular stenosis occur simultaneously and as a consequence, the continuity equation cannot be used [97]. In these cases, the planimetry method is recommended using direct aortic valve opening area tracing, as it is less flow dependent [97]. The accuracy of planimetry is much better using transesophageal echocardiography (TEE) when the imaging plane is placed at the level of the leaflet tips [97]. Furthermore, three-dimensional (3D) TEE proved superior accuracy when compared with 2D TEE, as a plane adjustment on 3D TEE can allow a true “en face” view aligned exactly to the smallest stenotic orifice.

The discordance between echocardiography parameters is most commonly observed in the case of depressed left ventricular systolic function (also known as classic low-flow, low-gradient aortic stenosis with reduced ejection fraction subtype) [97]. Patients with preserved left ventricular systolic function and tight aortic valve opening area, but low-gradients (also known as low-flow, low-gradient aortic stenosis with preserved ejection fraction subtype), represent another diagnostic issue [97]. According to current guidelines, stress-echocardiography and computed tomography (CT) imaging are recommended when the echocardiography is inconclusive [7]. The evaluation of aortic valve calcium by Agatson Ca-score has proved an adequate association with the severity of the aortic valve disease [98,99]. Simard and coworkers showed that women had less valvular calcification, but more fibrosis compared with men with similar aortic stenosis severity [100]. The CT-derived aortic valve Ca-score has become an important clinical tool, as it can identify severe aortic valve stenosis with a calcium score of ≥1600 Agatson unit (AU) in women and ≥3000 AU in men as very likely, ≥1300 AU in women and ≥2000 AU in men as likely, and if <800 AU in women and <1600 AU in men, then it is unlikely [98]. Furthermore, quantification of valve calcification may predict disease progression and major adverse events [101]. Despite the advantage of hemodynamic independence, Ca-score assessment in mainly fibrotic valves (e.g., bicuspid valves) is less useful, as fibrotic thickening can also determine an aortic valve opening area. Multi-slice cardiac CT also provides accurate measurements of the aortic valve opening area [102]. Yura Ahn and coworkers examined the CT characteristics of aortic stenosis and compared the aortic valve opening area measured by CT and echocardiography in 511 patients with different subtypes of aortic stenosis [103]. The two modalities showed high concordance (89%) to classify severe aortic stenosis, however, 56 patients were reclassified as moderate aortic stenosis by CT evaluation [103].

Aortic valve stenosis results in pressure overload and consequent myocyte enlargement and concentric left ventricular hypertrophy. In the early reactive phase, the myocardial fibrosis is diffuse and reversible if an aortic valve replacement is performed [104]. Otherwise, the persistence of pressure overload leads to myocyte apoptosis and focal, substitutive myocardial fibrosis that is usually irreversible and associated with adverse outcomes [105]. Myocardial fibrosis is best detected by cardiac magnetic resonance (CMR) using late gadolinium enhancement imaging and myocardial T1 mapping. Previous studies revealed that approximately 16% of patients with severe aortic stenosis have transthyretin cardiac amyloidosis manifesting in a more severe, progressive disease [106,107]. Transthyretin is a transport protein that transports the thyroid hormone thyroxine and retinol to the liver. The misfolded form of transthyretin can form amyloid deposits in the heart, including the myocardium and the valves. Despite both aortic stenosis and transthyretin cardiac amyloidosis being age-related diseases, the precise pathophysiological association between them is not well understood. Other than invasive myocardial biopsy and bone scintigraphy diagnostic tools, CMR has proved to be a good alternative noninvasive diagnostic modality in amyloidosis [107,108]. Furthermore, the special decreased longitudinal strain pattern with apical sparing on the speckle tracking echocardiography image, as a hallmark of cardiac amyloidosis, may raise the suspicion of the disease. Thickening of the valves, interatrial septum and ventricular wall with myocardial granular sparkling and minimal pericardial effusion are further echocardiographic characteristics associated with cardiac amyloidosis [107,109]. Castaño and coworkers found that patients with concurrent aortic valve stenosis and transthyretin amyloidosis had a 56% one-year all-cause mortality compared with 20% of patients with isolated aortic stenosis [110].

In summary, evaluation of the severity of an aortic valve stenosis is crucial for decision-making regarding patient management and the timing of intervention. Concordance of aortic stenosis with other diseases, such as hypertrophic cardiomyopathy or amyloidosis, may complicate diagnosis and alter the prognosis of the patient. Echocardiography is indicated for diagnosis in patients with symptoms of aortic stenosis, usually at a later stage of the disease. Even severe aortic stenosis can remain asymptomatic for a long period of time. Routine screening with echocardiography examination at an older age could be a solution, however, it would result in a significant economic and public health burden. To date, there are no diagnostic screening tools or algorithms in clinical practice to detect early stages of the aortic valve disease [68]. Future sensitive and specific molecular and imaging diagnostic tools could help to identify the early stages of the disease prior to irreversible macroscopic, later stage valve calcification.

## 7. Unmet Need for Effective Medical Therapy in Aortic Valve Stenosis

According to the current guidelines, aortic valve replacement is an effective treatment for severe aortic valve stenosis. This includes surgical valve replacement, however, less invasive transcatheter aortic valve replacement is performed in patients with high or intermediate surgical risk [7,111,112]. Nonetheless, the population in need of aortic valve replacement is rising with an aging society and increasing life expectancy. The increasing number of procedures represents an evolving economic and public health burden. Consequently, there is an unmet need for new effective medical treatments to at least halt the progression of the disease. Treatment of risk factors for atherosclerotic coronary artery disease, such as hyperlipidemia, hypertension, diabetes mellitus and smoking, reduces vascular atherosclerosis progression and mortality [113]. Despite these risk factors overlapping with aortic stenosis risk factors, their treatment has little effect on the progression of aortic stenosis.

Statins influence risk factors for atherosclerosis and modify inflammatory pathways by lowering lipid levels and exerting anti-inflammatory effects. Previous randomized trials examining the effect of lipid lowering therapy failed to prove a reduction in the progression of aortic valve stenosis and therefore, the guidelines do not recommend statin treatment to slow the progression of aortic valve calcification [84,114]. The ASTRONOMER and SALTIRE trials assessed the effect of intensive cholesterol lowering therapy on the progression of aortic valve stenosis [83,115]. Administration of rosuvastatin 40 mg or atorvastatin 80 mg did not halt the progression of aortic stenosis [83,115]. The large SEAS randomized trial revealed a reduction in ischemic events, but there was no difference in the progression of aortic stenosis with the use of 40 mg simvastatin plus 10 mg ezetimibe over 4 years’ follow-up [84,116]. A more targeted lipid lowering therapy, such as Lp(a)-lowering therapies would be of interest in the future, as the existing lipid-modifying drugs have modest effects on circulating Lp(a) levels [117]. Proprotein convertase, subtilisin/kexin type 9 (PCSK9) and Lp(a) inhibitors reduce non-Lp(a) and Lp(a)-containing lipoprotein particles, respectively [68]. It is known that ribonucleic acid inhibitor (RNAi) drugs result in temporary and reversible down-regulation of Lp(a) gene expression [118]. Preclinical and early clinical studies suggest that RNAi drugs may provide effective management of Lp(a)-mediated diseases [119,120]. The failure of statins in aortic stenosis might be explained by the pathophysiological course of the disease. Lipid deposition and inflammation are crucial in the initiation phase, however, aortic valve disease is diagnosed only in the later propagation phase. In the late phase of the disease, a self-perpetuating circle of calcium deposition with valvular injury is present and lipid deposition already has little impact [101]. Hypertension may initiate valvular injury in both phases of the disease. Antihypertensive therapy is the only medical therapy recommended by the guidelines for aortic stenosis accompanied by hypertension. Blood pressure control can avoid the additive effect of hypertension on the progression of aortic valve stenosis and afterload increases the result in further left ventricular hypertrophy [114]. Beyond their antihypertensive effect, renin-angiotensin-aldosterone system inhibitors may have an antifibrotic effect on the aortic valve and myocardium by reducing the expression of IL-6 [68,121]. Anti-inflammatory therapies still remain a question since no randomized trials have evaluated the effect of these drugs on aortic valve calcification [9].

Senescent cells can accumulate due to persistent mechanical stress on the aortic valve and cell cycle exit, contributing to valve degeneration. These cells secrete proinflammatory cytokines, matrix metalloproteinases and growth factors with additional senescence-promoting “bystander effect” on the surrounding cells [41,122]. Furthermore, senescent cells can impair the regenerative capacity of progenitor cells and, consequently, accelerate disease progression [38]. Apoptosis induction and removal of senescent cells from aortic valve tissue with senolytics could potentially halt the progression of the disease [37,41,94,123]. Targeting anti-apoptotic pathways of senescent cells can be another potential medical therapy termed senolytics [123]. Zhu and coworkers demonstrated that siRNA-mediated inhibition of anti-apoptotic pathways of senescent cells can induce apoptosis of these cells, but not of proliferating or quiescent cells [123]. Senolytics, such as dasatinib (FDA-approved tyrosine kinase inhibitor), quercetin (flavonoid) and navitoclax, can promote apoptosis of senescent cells [41,124]. The body of literature showed that the elimination of senescent cells can delay and prevent cardiovascular diseases, however, the benefit in clinical use remains to be proven [41,124,125,126]. Treatments focusing on the propagation phase and on breaking the self-perpetuating circle of valvular injury and calcium deposition with osteogenic differentiation might represent a future therapeutic strategy. However, bisphosphonates and denosumab, used in osteoporosis treatment, failed to affect the progression of aortic valve calcification [127].

## 8. Conclusions

Aortic valve stenosis is the most common valvular heart disease. The prevalence of aortic valve stenosis is increasing with an aging society and increasing life expectancy, representing an evolving economic and public health burden. Despite that it was thought to be a passive, age-related process, aortic stenosis is not an inevitable consequence of aging. Although the implication of age-related, replicative cellular senescence is not negligible, stress-induced cellular senescence plays a more important role in aortic valve calcification. Furthermore, aortic valve stenosis encompasses several active processes. The early initiation and the later progression phase are the two stages of aortic valve calcification that lead to severe aortic valve stenosis and heart failure. The early phase of aortic valve degeneration is dominated by inflammation, subendothelial lipoprotein oxidation, fibrosis and microcalcification, while the later phase is dominated by self-perpetuating progressive calcification [9,10,11]. The pathomechanism of aortic valve sclerosis and vascular atherosclerosis share many similarities, however, the progression phase is much different [93]. Vascular atherosclerosis can progress to plaque instability and rupture, while aortic valve calcification is a stable progressive process with lamellar bone formation [93]. Senescence cells secrete factors promoting vascular plaque instability. However, the impact of senescence cells is different in valves, as they lead to extracellular matrix remodeling. [94]. To date, no effective medical therapy is available to halt disease progression [7]. Surgical repair or replacement of the aortic valve represents the current standard of care for severe aortic stenosis [7]. A more detailed understanding of the stages of the aortic valve calcification could lead to novel treatment options to halt disease progression.

## Data Availability

Not applicable.

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
