# Peer review of "Cellular Senescence, Aging and Non-Aging Processes in Calcified Aortic Valve Stenosis: From Bench-Side to Bedside"

_cells, 2022, doi:10.3390/cells11213389_

Round 1
Reviewer 1 Report
In this review the authors try to summarize the pathophysiology of aortic stenosis and the clinical aspects of the disease.
The review is well written, however I would suggest some minor improvements
1) The immune system role in AS: Immune cells can remove senescent cells and prevent their accumulation, except in case of increased stresses-induced cellular senescence and immune system dysfunction. Is there immune system dysfunction in AS?
2) The authors suggest that The chronic mechanical stress over years results in the accumulation of senescent endothelial cells, even in case of normal laminar blood flow and mainly on the aortic side of the valve- what is the mechanism? And why does it happen only in 4-5% of population
3) The issue of ATTR and AS is interesting, however it does not add to the topic of the review.
Author Response
Response to Reviewer 1 Comments
We would like to express our thanks to Reviewer#1 for the careful evaluation of the manuscript and the helpful and constructive suggestions. We have heeded the Reviewer’s helpful propositions and prepared a revised version of the manuscript, which includes the alterations suggested by the Reviewer. Please find our point-by-point responses to the comments below.
Point 1: The immune system role in AS: Immune cells can remove senescent cells and prevent their accumulation, except in case of increased stresses-induced cellular senescence and immune system dysfunction. Is there immune system dysfunction in AS?
Response 1: Thank you for your important point! Aging is a complex process including age-related, physiologic, and replicative cellular senescence. These cells are eliminated by the immune system (Song P et al 2020). Further pathologic cellular senescence may occur in the presence of risk factors (hypertension, hypercholesterinaemia, diabetes mellitus, chronic renal disease) leading to mechanical, metabolic, and oxidative stress (Driscoll K et al 2021). It is supposed, that the immune system can not combat the clearance of these pathologic senescent cells leading to the accumulation of these cells (Song P et al 2020, Zhang L et al, 2022). Notably, the precise mechanism underlying senescent cell accumulation within tissues is still debatable (Song P et al 2020). However, the immune cells also become senescent with aging (Song P et al 2020, Zhang L et al 2022). Many functions of the immune system progressively decline with age, which further aggravates the rate of senescent cell accumulation (Song P et al 2020). Cellular senescence is part of the aortic valve calcification process, however, the presence of cardiovascular risk factors can aggravate valvular calcification and lead to severe stenosis (Driscoll K et al 2021). Notably, calcified aortic valve stenosis is not defined as an immune disease. Nonetheless, the role of the immune system in aortic valve calcification is not negligible (Driscoll K et al 2021).
Point 2: The authors suggest that The chronic mechanical stress over years results in the accumulation of senescent endothelial cells, even in case of normal laminar blood flow and mainly on the aortic side of the valve- what is the mechanism?
Response 2: Thank you for your question! Indeed, the sentence is misleading. Accordingly, we modified the text of the manuscript (highlighted below). It is hypothesized, that aging alone might not lead to severe aortic valve calcification, as a significant proportion of the elderly does not develop severe aortic stenosis. In the presence of risk factors, accumulation of pathologic senescent cells could be detected, which correlated with the valve tissue remodeling severity suggesting the role of cellular senescence in the progression of aortic valve calcification (Oh KS et al 2021). Aortic valve calcification affects mainly the aortic side of the valve, where the blood flow is oscillatory (Driscoll K et al 2021, Dayawansa NH et al 2022 ). Oscillatory blood flow represents different mechanical stress compared to the normal velocity laminar blood flow (Driscoll K et al 2021, Dayawansa NH et al 2022). Oscillatory shear stress has been shown to promote atherosclerotic plaque formation in arteries, and this is also assumed to be the case in aortic valve calcification, as calcification typically presents on the aortic side of the valve (Cunningham KS et al 2005; Yip CYY et al, 2011)
And why does it happen only in 4-5% of population?
Previously, calcified aortic valve stenosis was thought to be a passive, age-related disease (Dayawansa NH et al 2022). Now, aortic valve calcification is believed to be an active, multifactorial process with complex pathomechanism, including aging, senescence, cardiovascular risk factors, inflammation, fibrosis, osteoblast differentiation, etc. (Driscoll K et al 2021, Dayawansa NH et al 2022). The prevalence of cardiovascular risk factors is higher compared to the prevalence of severe calcified aortic valve stenosis. The global prevalence of hypertension in adults is 32-34%, meanwhile, the prevalence of aortic valve sclerosis, which is the early phase of aortic valve calcification, is approximately 20-30% and the prevalence of significant stenosis is only 2-3% in elderly (NCD-RisC members 2021, Lindroos M et al 1993). These differences might be explained by the complex pathomechanism of aortic valve calcification. However, the overall public health burden of severe aortic stenosis is escalating rapidly due to the increasing lifespan and prevalence of risk factors (Yi B et al, 2021; Yadgir S et al, 2020).
Accordingly, we modified the text as highlighted below:
“Apart from physiologic, age-related senescence, further pathologic cellular senescence can be induced with cellular stressors, such as excessive mechanical stress, oxidative stress, metabolic stress, and factors leading to DNA damage, also termed as stress-induced premature cellular senescence [26,27]. The chronic excessive stress over years may result in the accumulation of pathologic senescent endothelial cells, mainly on the aortic side of the valve, where the blood flow is oscillatory. Notably, aging of the immune system, defined as immunosenescence, may result in declined clearance and accumulation of senescent cells [27]. However, aging rarely leads to severe aortic valve stenosis, as a significant proportion of the elderly population does not develop significant AS. In addition to aging, excessive mechanical stress, genetic factors, and metabolic factors, such as high blood pressure, overweight, and hypercholesterinemia can induce and aggravate pathological cell senescence and calcification [36,37]. Excessive mechanical stress is present in case of increasing aortic stenosis severity, when the blood flow is oscillatory at the aortic side and turbulent at the ventricular side. Oscillatory flow represents different mechanical stress compared to the laminar flow of the normal aortic valve. Oscillatory shear stress has been shown to promote atherosclerotic plaque formation in arteries, and this is also assumed to be the case in aortic valve calcification, as calcification typically presents on the aortic side of the valve [10]. Besides senescence, mechanical stress can lead to focal endothelial damage and denudation [38].”
Point 3: The issue of ATTR and AS is interesting, however it does not add to the topic of the review.
Response 3: Thank you for the comment! Both the bench side and bedside issues of aortic valve calcification are complex. Approximately 16% of patients with severe aortic stenosis have transthyretin cardiac amyloidosis manifesting in a more severe progressive disease (Penalver J et al 2020; Thaden J.J. et al 2014). Both aortic stenosis and transthyretin cardiac amyloidosis are age-related diseases, however, the precise pathophysiological association between them is not well understood. We intended to emphasize shortly the clinical importance and challenging diagnostic feature of cardiac amyloidosis in aortic valve stenosis.
References
Cunningham, K.S.; Gotlieb, A.I. The role of shear stress in the pathogenesis of atherosclerosis. Lab Invest 2005, 85, 9–23, doi:10.1038/labinvest.3700215.
Dayawansa, N.H.; Baratchi, S.; Peter, K. Uncoupling the Vicious Cycle of Mechanical Stress and Inflammation in Calcific Aortic Valve Disease. Front Cardiovasc Med 2022, 9, 783543, doi:10.3389/fcvm.2022.783543.
Driscoll, K.; Cruz, A.D.; Butcher, J.T. Inflammatory and Biomechanical Drivers of Endothelial-Interstitial Interactions in Calcific Aortic Valve Disease. Circ Res 2021, 128, 1344-1370, doi:10.1161/circresaha.121.318011.
Lindroos, M.; Kupari, M.; Heikkila, J. and Tilvis, R. Prevalence of aortic valve abnormalities in the elderly: an echocardiographic study of a random population sample. Journal of the American College of Cardiology 1993, 21 (5), 1220–1225.
NCD Risk Factor Collaboration (NCD-RisC) members. Worldwide trends in hypertension prevalence and progress in treatment and control from 1990 to 2019: a pooled analysis of 1201 population-representative studies with 104 million participants. Lancet 2021, 398, 957–80, doi.org/10.1016/S0140-6736(21)01330-1.
Oh, K.S.; Febres-Aldana, C.A.; Kuritzky, N.; Ujueta, F.; Arenas, I.A.; Sriganeshan, V.; Medina, A.M.; Poppiti, R. Cellular senescence evaluated by P16INK4a immunohistochemistry is a prevalent phenomenon in advanced calcific aortic valve disease. Cardiovasc Pathol 2021, 52, 107318, doi:10.1016/j.carpath.2021.107318.
Penalver, J.; Ambrosino, M.; Jeon, H.D.; Agrawal, A.; Kanjanahattakij, N.; Pitteloud, M.; Stempel, J.; Amanullah, A. Transthyretin Cardiac Amyloidosis and Aortic Stenosis: Connection and Therapeutic Implications. Curr Cardiol Rev 2020, 16, 221-230, doi:10.2174/1573403x15666190722154152.
Song, P.; An, J.; Zou, M.H. Immune Clearance of Senescent Cells to Combat Ageing and Chronic Diseases. Cells 2020, 9, doi:10.3390/cells9030671.
Thaden, J.J.; Nkomo, V.T.; Enriquez-Sarano, M. The global burden of aortic stenosis. Prog Cardiovasc Dis 2014, 56, 565-571, doi:10.1016/j.pcad.2014.02.006.
Yadgir, S.; Johnson, C.O.; Aboyans, V.; Adebayo, O.M.; Adedoyin, R.A.; Afarideh, M.; Alahdab, F.; Alashi, A.; Alipour, V.; Arabloo, J.; et al. Global, Regional, and National Burden of Calcific Aortic Valve and Degenerative Mitral Valve Diseases, 1990-2017. Circulation 2020, 141, 1670-1680, doi:10.1161/circulationaha.119.043391.
Yi, B.; Zeng, W.; Lv, L.; Hua, P. Changing epidemiology of calcific aortic valve disease: 30-year trends of incidence, prevalence, and deaths across 204 countries and territories. Aging (Albany NY) 2021, 13, 12710-12732, doi:10.18632/aging.202942.
Yip, C.Y.Y.; Simmons, C.A. The aortic valve microenvironment and its role in calcific aortic valve disease. Cardiovasc Pathol Off J Soc Cardiovasc Pathol 2011, 20, 177–182, doi:10.1016/j.carpath.2010.12.001.
Zhang, L.; Pitcher, L.E.; Yousefzadeh, M.J; Niedernhofer, L.J.; Robbins, P.D.; Zhu, Y. Cellular senescence: a key therapeutic target in aging and diseases. J Clin Invest 2022, 132(15): e158450.
Once again, we would like to thank the Reviewer for the insightful comments and suggestions! We do believe these resulted in a much improved manuscript which may be acceptable for publication.
Andrea Agnes Molnar, MD, PhD
corresponding author

Reviewer 2 Report
The prepared review “Cellular senescence, aging and non-aging processes in aortic valve stenosis: from bench-side to bedside” is devoted to calcified aortic stenosis and includes a detailed description of the anatomy of the aortic valve itself, pathogenetic mechanisms of calcification, risk factors for the formation of aortic stenosis. The review also compares the pathogenetic mechanisms of atherosclerosis development and calcified aortic stenosis. The authors describe diagnostic techniques that are used in the diagnosis of aortic stenosis, as well as the possibilities of medical therapy for the prevention and conservative treatment of this pathology. The presented review is very detailed, well structured, includes a description of modern approaches and views on calcified aortic stenosis, the authors used current literature sources. There are no principal comments to this article.
It is possible to recommend to the authors to emphasize in the title of the article that this work deals specifically with calcified aortic stenosis "Cellular senescence, aging and non-aging processes in calcified aortic valve stenosis: from bench-side to bedside”, since there are also other possible causes of the formation of aortic stenosis, such as сhronic rheumatic heart disease, bicuspid aortic valve, infectious endocarditis, but they are not mentioned in this article.
Author Response
Response to Reviewer 2 Comments
We would like to express our thanks to Reviewer#2 for the careful evaluation of the manuscript and constructive suggestion. We would also like to thank for the encouraging comments and appreciation of our work. Please find our response below.
Point 1: It is possible to recommend to the authors to emphasize in the title of the article that this work deals specifically with calcified aortic stenosis "Cellular senescence, aging and non-aging processes in calcified aortic valve stenosis: from bench-side to bedside”, since there are also other possible causes of the formation of aortic stenosis, such as сhronic rheumatic heart disease, bicuspid aortic valve, infectious endocarditis, but they are not mentioned in this article.
Response 1: Thank you for the suggestion! Accordingly, we have changed the title of the manuscript to "Cellular senescence, aging and non-aging processes in calcified aortic valve stenosis: from bench-side to bedside”.
Andrea Agnes Molnar, MD, PhD
corresponding author